# HPV16 Induces Formation of Virus-p62-PML Hybrid Bodies to Enable Infection

**DOI:** 10.3390/v14071478

**Published:** 2022-07-05

**Authors:** Linda Schweiger, Laura A. Lelieveld-Fast, Snježana Mikuličić, Johannes Strunk, Kirsten Freitag, Stefan Tenzer, Albrecht M. Clement, Luise Florin

**Affiliations:** 1Institute for Virology and Research Center for Immunotherapy (FZI), University Medical Center, Johannes Gutenberg University of Mainz, Obere Zahlbacher Strasse 67, 55131 Mainz, Germany; linda.schweiger@gmx.de (L.S.); laura.aline.fast@gmail.com (L.A.L.-F.); snjezana.mikulicic@gmail.com (S.M.); johannes.strunk@uni-mainz.de (J.S.); kfreitag@uni-mainz.de (K.F.); 2Institute of Immunology, University Medical Center, Johannes Gutenberg University of Mainz, 55131 Mainz, Germany; tenzer@uni-mainz.de; 3Institute of Pathobiochemistry, University Medical Center, Johannes Gutenberg University of Mainz, Duesbergweg 6, 55128 Mainz, Germany; clement@uni-mainz.de

**Keywords:** human papillomavirus, HPV16, L2, p62, sequestosome-1, autophagy, antiviral defense, promyelocytic leukemia nuclear bodies (PML NB), hybrid bodies

## Abstract

Human papillomaviruses (HPVs) inflict a significant burden on the human population. The clinical manifestations caused by high-risk HPV types are cancers at anogenital sites, including cervical cancer, as well as head and neck cancers. Host cell defense mechanisms such as autophagy are initiated upon HPV entry. At the same time, the virus modulates cellular antiviral processes and structures such as promyelocytic leukemia nuclear bodies (PML NBs) to enable infection. Here, we uncover the autophagy adaptor p62, also known as p62/sequestosome-1, as a novel proviral factor in infections by the high-risk HPV type 16 (HPV16). Proteomics, imaging and interaction studies of HPV16 pseudovirus-treated HeLa cells display that p62 is recruited to virus-filled endosomes, interacts with incoming capsids, and accompanies the virus to PML NBs, the sites of viral transcription and replication. Cellular depletion of p62 significantly decreased the delivery of HPV16 viral DNA to PML NBs and HPV16 infection rate. Moreover, the absence of p62 leads to an increase in the targeting of viral components to autophagic structures and enhanced degradation of the viral capsid protein L2. The proviral role of p62 and formation of virus-p62-PML hybrid bodies have also been observed in human primary keratinocytes, the HPV target cells. Together, these findings suggest the previously unrecognized virus-induced formation of p62-PML hybrid bodies as a viral mechanism to subvert the cellular antiviral defense, thus enabling viral gene expression.

## 1. Introduction

Human papillomaviruses (HPVs) are small DNA viruses that consist of a circular double-stranded DNA that is enclosed by an icosahedral capsid composed of the major capsid protein L1 and the minor capsid protein L2 [1,2]. HPVs are divided into low- and high-risk types [3]. Low-risk types cause benign warts, whereas high-risk types can cause anogenital and oropharyngeal cancer. High-risk HPV type 16 (HPV16) is the most oncogenic HPV type and is the etiologic agent of cervical cancer [4,5]. Although vaccines against common tumorigenic HPV types have been established, the burden of HPV-associated cancer and disease remains high [6,7].

HPVs gain access to their host cells via lesions in the epithelium. After initial binding to primary attachment receptors, conformational changes of capsid proteins, and host cell signaling, virus particles are internalized via an unconventional clathrin-independent endocytic pathway [8,9,10,11]. HPV-containing endosomes undergo maturation to multivesicular endosomes, marked by a low pH, which aids in capsid disassembly and the translocation of L2 across the endosomal membrane whereby the L2 C-terminus faces the cytosol. L2 interaction with cytosolic proteins facilitate further virus transport via the retrograde pathway and along the microtubule network. With the onset of mitosis, the breakdown of the nuclear membrane enables the accumulation of viral DNA at promyelocytic leukemia nuclear bodies (PML NBs) [8,12,13].

PML NBs are dynamic cellular structures that dis- and re-assemble during mitosis [14]. The PML protein, also known as TRIM19, is the major component of PML NBs and plays important roles in both transcriptional regulation and antiviral activities [15,16]. The infectious complex, consisting of viral DNA (vDNA), L2 and a fraction of L1, remains in a membrane-bound transport vesicle until the completion of mitosis and the reformation of the nuclear envelope [17,18,19]. In the host cell nucleus, HPV modulates the composition of PML NBs, possibly to sustain a protective environment after disruption of the transport vesicle membrane to evade cellular defense mechanisms [13,20].

As additional antiviral defense, HPV16 can be routed to cellular degradation compartments such as autophagosomes or lysosomes [8]. At the same time, this virus has evolved ways to evade autophagy through activation of the mammalian target of the rapamycin (mTOR) pathway [21]. Autophagy-adaptor protein p62/SQSTM1 (SQSTM1 for sequestosome-1, hereafter p62) is one of the key factors in autophagy. This adaptor protein binds to ubiquitinated cargos and the microtubule-associated protein 1 light chain 3 (LC3) to form autophagosomes [22,23,24,25]. Aside from mediating autophagic degradation, p62 is also part of cellular signaling pathways affecting cell survival and oncogenesis [26,27]. p62 has recently been shown to sustain later stages of HPV infectious cycle by mediating degradation of the antiviral protein GATA-4 and thereby modulating viral replication [28].

In this study, we uncover the interaction of incoming HPV16 pseudoviruses (PsVs) with the autophagy adaptor protein p62. Virus entry into the host cell induces the recruitment of p62 to virus-filled endosomes and subsequently causes an almost complete co-translocation of p62 and the virally transduced DNA (vDNA) into the nucleus. Furthermore, nuclear p62-vDNA complexes fuse with PML NBs to form hybrid bodies. Loss of p62 resulted in increased degradation of capsid protein L2 and reduced delivery of vDNA to PML NBs along with diminished gene expression. Therefore, we are able to demonstrate that p62 not only supports viral entry but that it also functions as a component of the virus transcription and replication compartment.

## 2. Materials and Methods

### 2.1. Cell Lines and Pseudoviruses

Human cervical carcinoma cell line HeLa was purchased from the German Resource Center for Biological Material (DSMZ, Braunschweig, Germany) and cultivated in Dulbecco’s modified Eagle’s medium (Thermo Fisher Scientific, Waltham, MA, USA) supplemented with 10% fetal calf serum (FCS, Biochrom AG, Berlin, Germany), 1% modified Eagle’s medium non-essential amino acids (Thermo Fisher Scientific) and antibiotics (Fresenius Kabi, Bad Homburg vor der Hoehe, Germany) at 37 °C. Normal human epidermal keratinocytes (NHEK) were purchased from PromoCell (Heidelberg, Germany) and grown according to manufacturer’s instructions.

HPV16 pseudoviruses (PsVs) were produced as described previously [29,30]. Expression plasmid carrying codon-optimized L1 and L2 wt, p16SheLL [29], was co-transfected with a pcDNA3.1(+) luciferase reporter plasmid into HEK 293TT cells using polyethylenimine [31]. For detection of the viral DNA, EdU-modified PsVs were used. The labeling of the viral DNA with 5-ethynyl-2′-deoxyuridine (EdU, Click-iT AlexaFluor^®^ 488 Imaging Kit, Thermo Fisher Scientific) was performed by supplementing the growth medium with 20 μM EdU at six hours post transfection [32,33]. Then, 48 hours after transfection, cells were lysed, and the PsVs were purified by gradient centrifugation using OptiPrep (Sigma-Aldrich). Quantification of marker plasmid positive PsV (viral genome equivalents, vge) was performed by qPCR in an AB 7300 RT-PCR System as described previously [34]. To produce viruses harboring the LCR, the pcDNA3.1 luciferase reporter plasmid was replaced by the pGL4.20 luciferase vector (pGL4.20 HPV16 LCR) containing the HPV16 long control region (LCR) as initially described in [35].

### 2.2. Antibodies and Reagents

HPV L1-specific rabbit polyclonal antibody (pAb) K75, HPV L2-specific mouse monoclonal antibody (mAb) L2-1, and mouse monoclonal antibody 33L1-7 were described previously [36,37,38]. Mouse mAb p62 (ab56416), rabbit mAb PML (sc-5621) and mouse mAb β-actin (A5441) were purchased from Abcam (Cambridge, UK), Santa Cruz (Dallas, TX, USA) and Sigma-Aldrich (St. Louis, MO, USA), respectively. Mouse mAb LC3B (0260-100) was purchased from Nano Tools (Teningen, Germany). Rabbit polyclonal antibody recognizing endogenous p62 phosphorylated at Ser349 (#95697) was purchased from Cell Signaling (Leiden, Netherlands). Alexa-conjugated secondary antibodies were provided from Invitrogen (Carlsbad, CA, USA). Horseradish peroxidase-coupled (HRP) secondary antibodies for immunoblot were purchased from Jackson ImmunoResearch Europe Ltd. (Cambridgeshire, UK). Autophagy inducer rapamycin and autophagy inhibitor 3-Methyladenine (3-MA; M9281) were provided by Sigma-Aldrich. Autophagy inhibitor bafilomycin A1 (B-1080) was provided by LC Laboratories (Woburn, MA, USA).

For L2 overexpression, pUF3/hu16L2 codon-optimized HPV16 L2 expression vector with ampicillin resistance was used [30]. P62 knockdown: p62-specific siRNAs #1 (GGGUGCAAGAAGCCAUUUAdtdt), #2 (CAUAGGUCCCUGACAUUUAdtdt), #3 (GGAGGAUCCGAGUGUGAAUdtdt) and #4 (ACGUUUGCAUAGAGAGAAAdtdt) were purchased from Sigma Aldrich (St. Louis, MO, USA). HeLa and NHEK cells were transfected with these siRNAs or a pool of all four single siRNAs using RNAiMAX (Invitrogen, Carlsbad, CA, USA) according to manufacturer’s instructions. Subsequent experiments were performed 48 h after transfection.

### 2.3. Endosomal Preparation and Quantitative Mass Spectrometry

For the analysis, quantitative mass spectrometry data of endosomal preparations were analyzed as published in part already in [39]. Briefly, HeLa cells were either left untreated or were infected with HPV16 PsVs. Next, the cells were harvested, homogenized and a post-nuclear supernatant was prepared. The sucrose centrifugation gradient was prepared and samples were centrifuged for 90 min at 14,000× *g*. Thirteen fractions were collected. Early endosomal fractions were identified by immunoblotting using the endosomal marker Rab5. Protein digest of early endosomes and mass spectrometry analysis of tryptic peptides were performed as described in [39] and in Appendix A.

### 2.4. HPV PsV Infection Assay

HeLa or NHEK cells were grown in 24- or 48-well plates and incubated with siRNAs, plasmids, autophagy inducers and/or autophagy inhibitors. Cells were infected with approximately 100 luciferase vector-positive HPV16 PsVs per cell for 24 h as described [40]. HPV PsV infection assay based on luciferase expression and activity was performed as described earlier [34]. Luciferase activity was normalized to lactate dehydrogenase (LDH) (CytoTox 96 Non-Radioactive Cytotoxicity Assay, Promega, Madison, WI, USA) as cell viability control, and data were expressed as relative infection rate normalized to controls.

### 2.5. Immunofluorescence Microscopy

HeLa or NHEK cells were grown on glass cover slips in 12-well plates. After transfection with siRNAs or expression plasmids and/or incubation with PsVs, HeLa and NHEK cells were fixed and permeabilized with methanol and paraformaldehyde/Triton X-100, respectively. Cells were incubated with primary antibodies and species-specific Alexa-conjugated secondary antibodies. DNA staining was performed with Hoechst 33342 (Sigma-Aldrich) to visualize nuclei. Cover slips were mounted with Dako Fluorescent Mounting Medium (Dako, Carpinteria, CA, USA). Image acquisition was performed using a Zeiss Axiovert 200 M microscope fitted with a Plan-Apochromat 100×/1.4 Oil objective and Axiovision deconvolution software (Carl Zeiss, Jena, Germany). Quantification of co-localization was performed by analysis of at least 20 pictures per group using Co-localization Software 4.7 (Carl Zeiss).

### 2.6. Co-Immunoprecipitation Assay

Co-immunoprecipitation of L1 and L2 after 18 h of HPV16 PsV incubation using Dynabeads (M-280 sheep anti-mouse IgG, Thermo Fisher Scientific) was performed as described earlier [39]. Co-immunoprecipitation after L2 overexpression was performed by growing HeLa cells in 6 cm culture dishes overnight and transfecting cells with L2 expression plasmid using polyethylenimine, (PEI), produced as described earlier [34], for 24 h. Cells were lysed in MACS lysis buffer (MACS Miltenyi Biotec, Bergisch Gladbach, Germany), and co-immunoprecipitation was performed using A/G agarose beads (Santa Cruz) as described earlier [41].

### 2.7. Duolink Proximity Ligation Assay (PLA)

HeLa cells were grown on cover slips in 12-well plates overnight and incubated with HPV16 PsV for 24 h. They were fixed and permeabilized with methanol. Proximity ligation assay (PLA) was performed according to manufacturer’s instructions using Duolink^®^ In Situ Orange Starter Kit Mouse/Rabbit was purchased from Sigma-Aldrich and antibodies against p62 and PML. Cover slips were mounted with Duolink^®^ Mounting Media with DAPI included in the kit. Image acquisition was performed using a Zeiss Axiovert 200 M microscope equipped with a Plan-Apochromat 100×/1.4 Oil. Quantification of fluorescent pixels was performed using at least 20 pictures per group with ImageJ software.

### 2.8. Statistics

All experiments were reproduced at least three times (biological replicates) unless stated otherwise. Statistical analyses were performed with GraphPad Prism 9 for Windows (GraphPad Software, San Diego, CA, USA, www.graphpad.com, access date: 9 May 2022). Data were tested for normal distribution using the Shapiro–Wilk statistical assay. Normally distributed data were further analyzed using the two-tailed, unpaired (independent samples) *t* test or using the ordinary one-way ANOVA test for comparing two or more than two groups, respectively. Non-normally distributed data were analyzed using the Mann–Whitney test or by the non-parametric ANOVA (Kruskal–Wallis) test for comparing two or more than two groups, respectively. The ‘n’ for each presented analysis (stated in Figure legend) denotes the sample size. Differences between the groups were considered statistically significant when *p* ≤ 0.05, with the statistical significance marked in the graph (*p* ≤ 0.05 *, *p* ≤ 0.01 **, *p* ≤ 0.001 ***, not significant n.s.). 

## 3. Results

Despite recent advances in the study of HPV infectious entry, the cellular mechanisms enabling nuclear delivery of the viral genome and viral strategies that overcome the host defense machinery are not fully understood.

### 3.1. The Autophagy Adaptor p62 is Recruited to HPV16 Pseudovirus-Filled Endosomes and Interacts with HPV Capsid Proteins

To identify cellular proteins modulating HPV16 entry into epithelial cells, we performed proteome analysis of endosomal preparations of HeLa cells treated with HPV16 pseudoviruses (PsVs). Part of this analysis has been published before [39] (table with all candidates, Appendix A). HeLa cells were control-treated (non-infected, n.i.) or incubated with HPV16 PsVs for either four or seven hours. Endosomes were prepared using flotation centrifugation in a sucrose step gradient as described [42,43]. Proteins in endosomal fractions were subjected to tryptic digestion and quantitative analysis by liquid chromatography–mass spectrometry (qLC-MS). Each preparation was analyzed in five technical replicates. About 400 proteins were reproducibly detected in at least three out of five technical replicates (Appendix A). The HPV16 L1 protein was detected only in PsV-treated preparations, and its amount increased during the time course of infection (Figure 1 and Appendix A). The marker protein Rab5a was present in all early endosomal fractions in comparable amounts. The autophagy adaptor p62/sequestosome-1 was identified as a strongly enriched protein at four and seven hours post PsV infection (hpi) (Figure 1a and Appendix A), while no autophagosome marker (e.g., LC3) was detected (Appendix A). These data suggest that p62 is recruited to virus-filled endosomes. Supporting this notion, Western blot analysis of endosomal preparations displayed enhanced p62 levels in sucrose density fractions containing Rab5a and HPV compared to non-infected (n.i.) controls (Figure 1b). In addition, p62 co-localized with HPV major capsid protein L1 at vesicular structures (Figure 1c).

To assess the putative interaction of p62 and incoming HPV16, HeLa cells were incubated with HPV16 PsVs and lysed for co-immunoprecipitation. Whole cell lysates were incubated with either p62-specific or control antibody-coupled beads. Western blot analysis of precipitated proteins showed that HPV major capsid protein L1 as well as minor capsid protein L2 were co-precipitated with p62 (Figure 1d), suggesting complex formation of p62 with HPV during virus entry. It is known that HPV16 L1 and vDNA are protected in a transport vesicle and that L2 adopts a transmembranous topography during entry with the majority facing the cytosolic side for the interaction with host cell transport factors [44]. To test the possible direct interaction of L2 and p62, we performed co-immunoprecipitation of L2 in the absence of L1 or viral DNA in L2 overexpressing HeLa cells. Our co-precipitation analysis verified the interaction of p62 and L2 (Figure 1e). These data support the notion that p62 is recruited to virus-filled transport vesicles by interaction with L2.

### 3.2. HPV16 Induces p62-PML Nuclear Body Fusion

To follow the fate of the autophagy adaptor p62 and HPV16 at later stages of virus entry, we analyzed co-localization of p62 and the virally transduced DNA (vDNA) using 5-ethynyl-2′-deoxyuridine (EdU)-labeled and encapsidated DNA of HPV16 PsVs by immunofluorescence analysis at 24 h post PsV addition. vDNA and p62 co-localized in distinct structures of the nucleus indicating virus-induced translocation of p62 from the cytoplasm to nucleus, the site of HPV16 transcription and replication (Figure 2a). Double staining of PML as a marker for PML NBs and p62 at 24 h after HPV16 PsV addition displayed strong overlap of p62 with the center of each PML NB compared to non-infected cells (Figure 2b). Quantification revealed a ten-fold increase in co-localizing pixels when compared to non-infected controls (Figure 2c). It is of note that total p62 protein amounts were not affected by PsVs treatment (Appendix A). In addition, p62 phosphorylation at Serine 349, induced by E6 and E7 HPV16 oncoproteins [28], was not detectable during the time course of virus entry (Appendix A). As the enhanced p62-PML co-localization indicates fusion of p62-complexes and PML NBs, a Duolink^®^ proximity ligation assay (PLA) was performed to corroborate the spatial proximity of p62 and PML with a better resolution (Figure 2d,e). The bright fluorescent PLA spots form at sites where two different molecular entities are less than ≈40 nm apart, which should suffice for physical interaction or fusion [45,46]. The five-fold increase in PLA signals at 24 h after PsVs addition demonstrates that HPV16 PsV entry into HeLa cells induced a strong translocation of p62 into the host cell nucleus and formation of p62-PML hybrid bodies. Triple staining shows clear nuclear structures with a p62-vDNA core surrounded by a shell of PML (Figure 2f). 

### 3.3. P62 is a Proviral Factor of HPV16 PsV Infection

To investigate the function of p62 in HPV16 infection, we performed a PsV infection assay in p62-depleted cells using p62-specific siRNAs (Figure 3). First, p62 knockdown efficiency was tested by transfection of HeLa cells with four different p62 siRNAs or a pool thereof. The siRNA #1, #2, #3 and the pool of all four siRNAs displayed efficient p62 protein depletion (Figure 3a) and were used in the HPV16 PsV infection assay. All p62-specific siRNAs significantly reduced infection rate (Figure 3b). Control experiments excluded an effect of p62 on virus internalization into the cells, which might have been responsible for the reduced infection rated (Appendix A). These findings uncover p62 as a novel proviral factor in HPV16 infection of HeLa cells.

### 3.4. P62 Protects HPV16 PsV from Autophagic Degradation and Supports Delivery to PML NBs

Since p62 is primarily acknowledged for its role in the degradation of autophagic cargos [22,23]; the role of autophagy in HPV16 infection was assessed. First, we overexpressed LC3B-GFP as a specific marker for autophagosomes and detected slight overlap of capsid protein L1 and LC3B-GFP (Appendix A). When the co-localization of LC3B-GFP and L1 was analyzed at different times post infection, no significant changes in LC3B-GFP-L1 co-localization were detectable (Appendix A). These data suggest a continuous flux of incoming PsVs via the autophagy pathway.

Host autophagy has been reported to inhibit HPV16 infection in primary keratinocytes [47]. To investigate the effect of autophagic processes on HPV16 infection in HeLa cells, we modulated autophagy by incubation with autophagy inhibitor 3-Methyladenine (3-MA) or autophagy inducer rapamycin and assessed HPV16 PsV infection rates in our cellular system. In line with a previous publication [21], PsV infection assays revealed that the relative infection rate was significantly increased in autophagy-inhibited cells and significantly decreased after autophagy stimulation (Figure 4a), supporting the notion that autophagy performs an antiviral function during HPV16 entry. 

Next, we assessed the role of p62 in L2 degradation (Figure 4b,c) and analyzed the fate of the viral genome (vDNA) in p62-depleted cells (Figure 4d–g) using HPV16 EdU PsV. For these experiments, HeLa cells were treated with p62-specific siRNA and incubated with HPV16 PsVs. Cells were additionally treated with bafilomycin A1 to block protein degradation, including the autolysosomal pathway (Figure 4b–e). The increase in p62, a well-known autophagy cargo, was used as a positive control for autolysosome inhibition upon bafilomycin A1 treatment (Figure 4b). In the absence of bafilomycin A1 treatment, Western blot analyses showed significantly decreased L2 levels in p62 siRNA-treated cells compared to control cells (Figure 4b,c). Bafilomycin A1 treatment not only prevented this p62-dependant reduction in L2 protein amount but also significantly increased L2 levels of control or p62 siRNA-treated cells to comparable levels. These data support the notion of p62 protecting L2 from degradation, most likely via the autophagosomal pathway, as the autophagic flux is blocked regardless of the presence or absence of p62.

Because part of the L1 protein is degraded by sorting into lysosomes after capsid disassembly [48], we analyzed the fate of the viral genome in p62-depleted cells at different time points after virus addition (8 hpi for cytoplasmic PsV staining and 24 hpi for analysis of PML-virus co-localization as shown before [20,39,40,48,49]). After incubation with PsVs, the cells were additionally treated with bafilomycin A1 to improve visualization of autophagosome marker LC3B prior to autophagosomal maturation to autophagolysosomes and its degradation. Imaging analysis revealed that co-localization of vDNA and LC3B significantly increased in p62 siRNA- and bafilomycin A1-treated cells (Figure 4d,e). This finding suggests that in the absence of p62, viral particles were routed to the autophagic machinery, whereas in the presence of p62, the virus subverts autophagic degradation, enabling successful delivery of the infectious complex to PML NBs. p62-depletion significantly reduced vDNA-PML co-localization (Figure 4f,g), while cellular PML protein amounts remained unaffected by p62 siRNA treatment (Appendix A).

### 3.5. HPV16 Induced Formation of Virus Containing p62-PML Hybrid Bodies Is Required for Gene Expression in Primary Keratinocytes

To control the relevance of our findings for a natural HPV16 infection, we reproduced key experiments in normal human epithelial keratinocytes (NHEKs), the primary target cell of HPV16. Triple staining of HPV16 PsV-incubated NHEKs verified vDNA-p62 co-localization within the cytoplasm after virus entry as well as the formation of vDNA-p62-PML hybrid bodies in the nucleus (Figure 5a). Quantification of co-localizing p62 and PML pixels showed virus-induced fusion of vDNA-p62 complexes with PML NBs at 24 h post virus addition (Figure 5b). These hybrid bodies might represent viral transcription compartments as depletion of p62 (Figure 5c), again leading to significantly decreased relative infection rate (Figure 5d). Here, we used PsVs with the marker gene luciferase under control of the HPV16 long control region containing the HPV16 viral early promoter. This approach combines the HPV16 target cells (primary keratinocytes, NHEK) with an infection system that is even more similar to natural HPV16 infection [35]. Western blot shows that only siRNA#3 and the pool of all four siRNAs were able to reduce the p62 levels in NHEK (Figure 5c) and the corresponding infection rates (Figure 5d). Our findings demonstrate that relative infection rates correlated with the p62 protein levels and reiterate that p62 plays a proviral role in HPV16 infections.

## 4. Discussion

Viruses have evolved strategies to escape the cellular immune defense by hijacking key host cell factors and by manipulation of antiviral processes [16,50,51]. In this report, we reveal that entry of the oncogenic human papillomavirus type 16 into epithelial cells induces translocation of the autophagy cargo receptor p62 into the nucleus and its incorporation into viral DNA-containing PML nuclear bodies. We show that p62 forms a complex with incoming viruses at vesicular structures most likely via the minor capsid protein L2. After p62 nuclear translocation, the p62-embedded viral DNA fuses with PML NBs, forming hybrid bodies that enable efficient viral gene transcription. Thereby, we unraveled a new function of p62 in the entry of HPV16 and a thus far unknown immune evasion strategy of an incoming virus. The findings obtained here also reveal HPV16 PsVs as a tool to further investigate the understudied role of nuclear p62 and of p62-PML hybrid bodies.

Using proteomics, we found that p62 increasingly accumulates at virus-filled endosomes during virus entry, suggesting complex formation of this autophagy adaptor protein with incoming virus particles. Our co-immunoprecipitation studies indicate a physical link between p62 and HPV16 capsid proteins L1 and L2. As only the C-terminal part of L2 penetrates the limiting endosomal membrane while L1 and the viral DNA remain in a transport vesicle, it seems plausible that L2 forms the link between p62 and the virus. Interaction of L2 and p62 has been confirmed in the absence of L1, supporting the notion that p62 is recruited to incoming HPV16 by interacting with L2. This interaction might be mediated by ubiquitin, as p62 is known to bind polyubiquitinated proteins [22,23,24], and L2 has been shown to be polyubiquitinated after expression as well as during the entry pathway [52,53]. It has been described that the loss of p62 has little effect on autophagy, although p62 recruits ubiquitinated cargos [22,54]. This lack of phenotypic impact has been explained by the presence of other autophagy adaptors such as Optineurin, Neighbor of BRCA1 gene 1 (NBR1), and nuclear dot protein 52 kDa (NDP52) which could take over p62 function [23]. However, p62 has multiple additional cellular roles [54]. During HPV16 entry, we have found that loss of p62 rather increases the degradation of the L2 capsid protein and enhances the co-localization of viral DNA with LC3B, suggesting that p62 plays a minor role as an autophagy receptor during this process. In HPV16 infection, p62 might compete with other autophagy adaptors for binding to L2. In addition, the interaction between incoming viruses and p62 could lead to masking of the LC3 interaction domain within the p62 protein sequence and thereby to the inhibition of autophagosomal sorting. Further analyses are necessary to elucidate this mechanism.

Complex formation of p62 and the virus was not only detected in the cytoplasm but also in the nucleus at PML NBs, suggesting co-transport and inclusion of these large complexes into newly formed and modified nuclear structures, the sites of HPV transcription and replication. Our imaging analyses visualize virally transduced DNA surrounded by an inner layer of p62 and an outer layer of PML proteins. Fusion of nuclear p62 bodies with PML bodies has been uncovered by electron micrographs under conditions when nuclear export of p62 was inhibited [55]. In addition, a fluid model proposed that p62 forms cytosolic and nuclear compartments delimited by phase separation containing free-floating ubiquitinated cargo [56,57]. These so-called p62 bodies allow included proteins to keep their conformation, enabling interaction and mutual influence. Therefore, we propose a model (shown in Figure 6) in which p62 is recruited to virus-harboring vesicles by interaction with L2. Subsequently, HPV16-containing p62 bodies fuse with PML NBs after completion of mitosis, thereby forming virus-p62-PML hybrid bodies. With these analyses, we also identified p62 as a thus far unknown component of the HPV16 transcription compartment.

In contrast to the assumption that p62 enables host cell defense by feeding HPV16 into autophagosomes, we observed that p62 acts as a proviral in HPV16 infection, as siRNA-mediated p62 depletion in HeLa cells and primary keratinocytes led to decreased infection rates of HPV16 PsVs. Considering the well-established role of p62 as an autophagy cargo receptor [22,23,24,25] and the reported HPV16-induced autophagosome formation [47,58,59], we analyzed the role of p62 during HPV16 entry, focusing on autophagy processes. First, we verified the earlier reported antiviral function of autophagy in our cell system. Induction of autophagy led to decreased infection rates, while inhibition had the opposite effect. Further analyses of HPV16 L2 protein degradation uncovered that p62 protects L2 from degradation. Likewise, vDNA was routed to autophagosomal structures in the absence of p62, which simultaneously resulted in a loss of vDNA in PML NBs. In an earlier study, it has been shown that p62 depletion in HeLa cells neither affects the number of autophagosomes nor autophagic turnover [60]. In a parallel approach, we identified no effect of p62 depletion on nuclear PML, excluding side effects on PML NBs. It is therefore plausible to hypothesize that p62 is not recruited to HPV-filled endosomes to mark them for autophagosomal degradation but is being hijacked by incoming HPV16 to enable both the evasion of the antiviral defense and the nuclear delivery of incoming viral genomes.

In this scenario, HPV16-induced signal transduction might contribute to p62 phosphorylation and its translocation into the nucleus as described during vaccinia virus infection [61]. Here, p62 is phosphorylated by a vaccinia virus kinase that is not present in papillomaviruses. Moreover, vaccinia- and papillomaviruses oppose the p62 requirement for infection, as p62 restricts vaccinia while supporting HPV16 infection. In the context of HPV16 infection, it has been shown that HPV16 oncogene-mediated activation of the ATR/p62 autophagy-related pathway results in p62 phosphorylation at serine 349 [28]. In contrast to these findings, we detected no changes in p62 Ser349 phosphorylation at any tested time point. However, it is possible that p62 phosphorylation at or near the nuclear localization signal contributes to the observed massive p62 nuclear translocation as shown before [62]. Further studies are required to uncover these underlying mechanisms.

In addition, the role of nuclear p62 is understudied. Although association of virus components with nuclear and PML resident p62 has recently been observed during human cytomegalovirus (HCMV) morphogenesis [63]; the role of this association has not been further analyzed. Likewise, the interaction between p62 and PML has been described in herpes simplex virus type 1 (HSV-1) infection [64]. Since PML restricts HSV-1 infection, p62 recruitment was proposed to mediate autophagosomal degradation of PML. During HPV16 infection, in addition to its protective role against autophagic degradation, nuclear p62 might contribute to the degradation of nuclear restriction factors such as MYPOP or Sp100 [35,65]. This hypothesis is supported by studies demonstrating that nuclear p62 aggregates serve as efficient centers for proteolysis of nuclear proteins or mediate nuclear export of the fusion protein PML-RARα, directing it toward autophagosomal degradation [57,66]. Further investigation will be needed to examine the role of nuclear p62 in HPV infection or in infections by other viruses that target PML NBs to create a favorable environment for the establishment of viral gene transcription and replication.

## Figures and Tables

**Figure 1 viruses-14-01478-f001:**
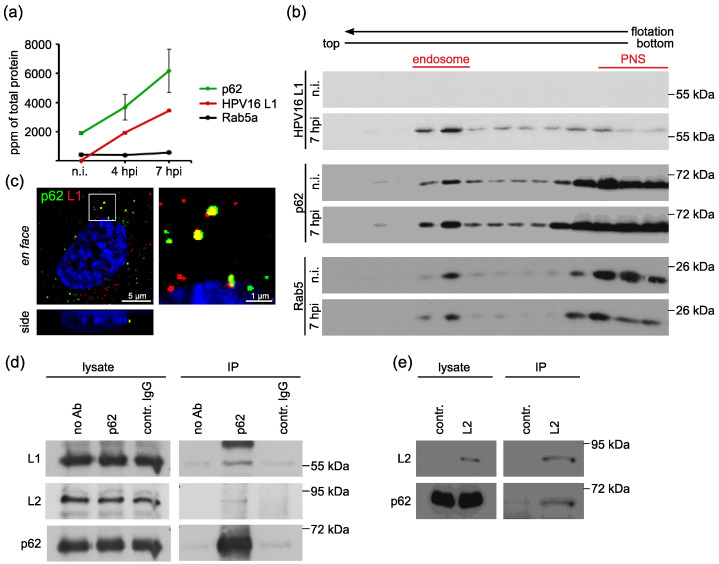
P62 is recruited to HPV16 PsV-filled endosomes and forms a complex with incoming PsVs and overexpressed L2. (**a**,**b**) Endosomal p62 level increases post PsV addition. HeLa cells were incubated with PsVs for 4 or 7 h (hours post infection, hpi) or left untreated (non-infected, n.i.), and endosomes are isolated by sucrose gradient centrifugation. (**a**) Parts per million (ppm) values of total protein for p62, HPV16 L1, and Rab5a protein of endosomal fraction determined by label-free quantitative mass spectrometry (see also Appendix A). The graph shows the mean of two independent experiments. Error bars show mean value ± standard error of the mean (SEM). (**b**) Western blot of HPV16 L1, p62 and Rab5 in sucrose gradient fractions. Rab5 serves as a marker for early endosomes. Endosomal fraction marked in Western blot was used in mass spectrometry. Western blot fractions marked with PNS represent post-nuclear supernatant rich in soluble cytoplasmic proteins such as Rab5. (**c**) Representative picture of p62 (green) and L1 (red) co-localization in HeLa cells. HeLa cells were incubated with PsVs for 7 h and stained with p62-specific mAb and L1-specific pAb K75. The box in the left panel marks the outline of p62 and L1 co-localization shown in higher magnification in the right panel. The picture shows both en face (xy) and side view (xz). The side view was generated from z-stacks. Nuclei are stained in blue (Hoechst 33342). (**d**) Co-immunoprecipitation of p62 and HPV16 capsid proteins. HeLa cells were incubated with PsVs for 18 h. Cells were lysed and incubated with p62-specific mAb for immunoprecipitation. Protein input was verified by Western blotting of whole cell lysates using specific antibodies. P62 immunoprecipitation and L1 and L2 co-immunoprecipitation were detected by Western blotting using p62- (mAb), L1- (pAb K75) or L2- (mAb L2-1) specific antibodies. (**e**) Co-immunoprecipitation of HPV16 minor capsid protein L2 and p62. HeLa cells were transfected with empty control or L2 plasmid for 24 h. Protein input was verified by Western blot of whole cell lysates using specific antibodies. The p62 precipitations and L2 co-immunoprecipitations were detected by Western blotting using anti-L2 or anti-p62 mAbs.

**Figure 2 viruses-14-01478-f002:**
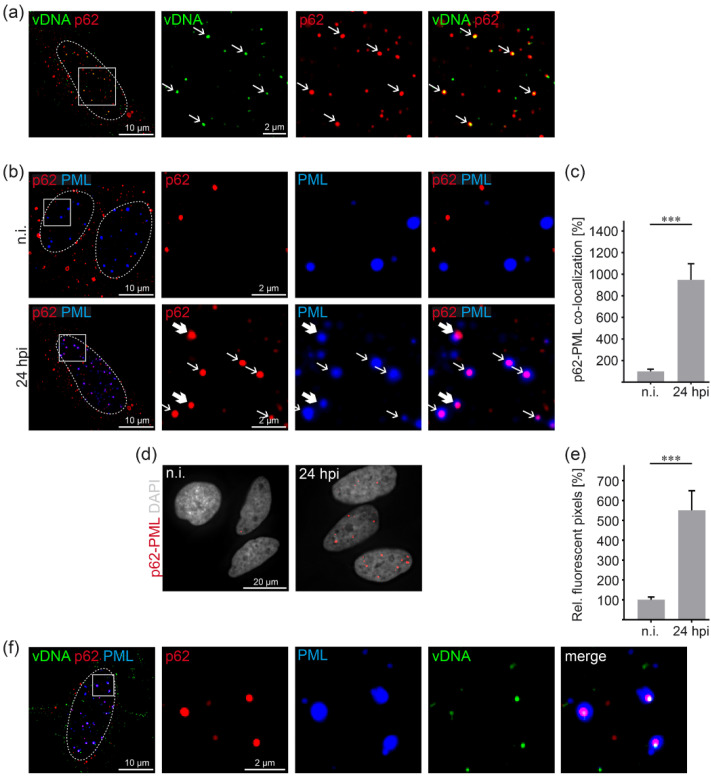
P62 and PsV transduced vDNA are co-transported to PML NBs and induce p62-PML nuclear body fusion. (**a**) Representative pictures of nuclear vDNA and p62 co-localization. HeLa cells were incubated with EdU-labeled PsVs for 24 h and stained using p62-specific mAb. The box in the first panel outlines the following higher magnification images showing the co-localization of vDNA (green) and p62 (red). Arrows mark co-localizing vDNA and p62. The dotted line depicts the nucleus. (**b**,**c**) PsV treatment of HeLa cells increases p62-PML co-localization. HeLa cells were incubated with EdU-labeled PsVs for 24 h (24 hpi) or left non-infected (n.i.) and stained with anti-p62 and anti-PML mAbs, respectively. (**b**) Representative pictures of p62 (red) and PML (blue) co-localization. Boxes on the left indicate areas of higher magnification in the following images, showing the co-localization of p62 and PML. Bold arrows mark the fusion onset of a p62 body with a PML body, whereas other arrows mark p62 bodies captured by PML NBs. (**c**) Relative co-localization of p62 and PML. p62 pixels co-localizing with PML pixels are shown as a mean ± SEM, and mean of non-infected cells (n.i.) was set to 100%. Data (*n* = 53–57 images of three replicates) were analyzed for significant differences using the Mann–Whitney test; *p* < 0.0001 (***). (**d**,**e**) Close association of p62 and PML increases after PsV infection. HeLa cells were incubated with HPV16 PsV for 24 h (24 hpi) or left non-infected (n.i.). Fusion of p62 and PML was analyzed by PLA. (**d**) Representative pictures showing fluorescent pixels (p62-PML, red) indicating close association (<40 nm) of p62 and PML in the nucleus (in grey, stained with Hoechst 33342). (**e**) Quantification of fluorescent pixels indicating close p62 and PML association per nuclear pixels. p62-PML positive pixels are shown as a mean ± SEM, and the mean for non-infected cells (n.i.) was set to 100%. Data (*n* = 60 images of three replicates) were analyzed for significant differences using Mann–Whitney test; *p* < 0.0001 (***). (**f**) Representative pictures of nuclear vDNA, p62 and PML co-localization. HeLa cells were incubated with EdU-labeled PsVs for 24 h and stained for immunofluorescence. The box area is enlarged and shows co-localization of p62 (red), PML (blue) and vDNA (green).

**Figure 3 viruses-14-01478-f003:**
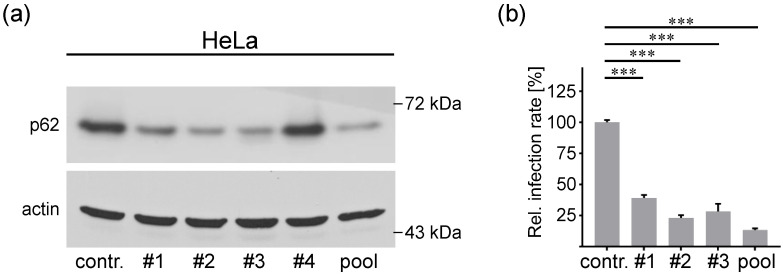
P62 is required for HPV16 PsV infection. (**a**) HeLa cells were transfected with control (contr.), p62-specific siRNAs (#1, #2, #3, #4) or p62 siRNA pool (pool) of all four siRNAs for 48 h. Knockdown efficiency was analyzed by Western blotting 48 h after siRNA transfection using p62-specific mAb. β-actin was used as loading control. (**b**) HeLa cells were transfected with p62-specific siRNAs for 48 h and infected with HPV16 PsVs for 24 h. Relative infection rate was assessed by luciferase activity and normalized to lactate dehydrogenase (LDH) activity as cell viability control. Infection rate is given as mean ± SEM, and the mean for control siRNA-treated cells was set to 100%. Data (*n* = 7 out of three replicates) were analyzed for significance using the one-way ANOVA test; *p* < 0.0001 (***).

**Figure 4 viruses-14-01478-f004:**
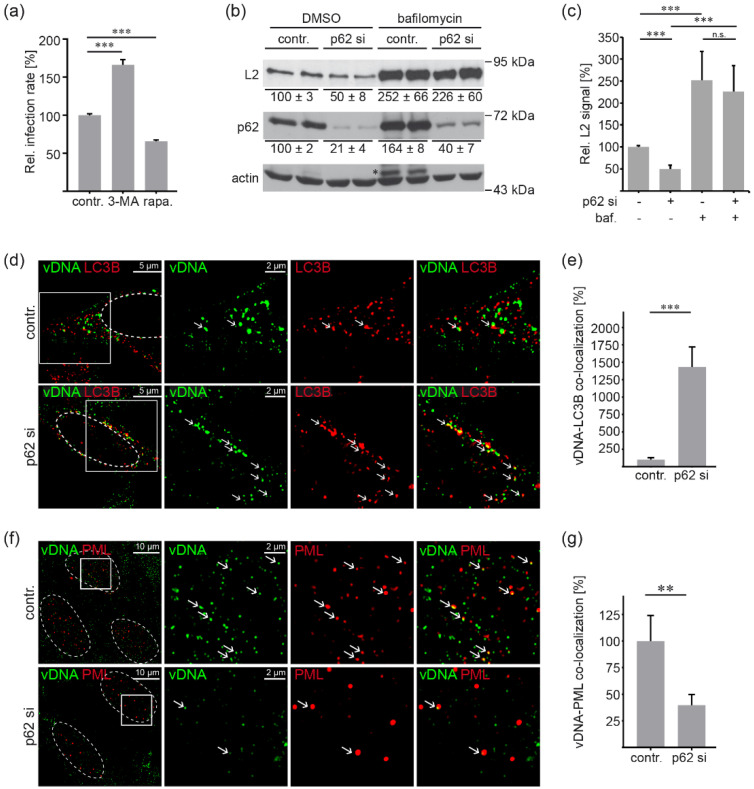
P62 protects HPV16 PsVs from autophagic degradation and supports delivery to PML NBs. (**a**) HPV16 infectivity is affected by autophagy up- and downregulation. HeLa cells were treated with autophagy inhibitor 3-Methyladenine (3-MA, 5 nM), autophagy-inducer rapamycin (rapa., 50 nM) or left untreated, and subsequently infected with HPV16 PsVs for 24 h. Relative infection rate was assessed by luciferase activity and normalized to LDH activity as cell viability control. Infection rate is given as a mean ± SEM, and the mean for control siRNA-treated cells (contr.) was set to 100%. Data (*n* = 15–32 out of three replicates) were analyzed for significant differences using non-parametric ANOVA (Kruskal–Wallis test): *p* < 0.0001, with contr. vs. 3-MA *p* = 0.0009 (3-MA) and contr. vs. rapa. *p* < 0.0001. (**b**,**c**) p62 protects the minor capsid protein L2 from degradation. HeLa cells were transfected with control (contr.) or a p62 siRNA pool (p62 si) for 48 h and incubated with HPV16 PsVs for 24 h either with or without bafilomycin A1 (baf. 1.25 µM). (**b**) Knockdown efficiencies of p62 and L2 viral protein were analyzed by Western blotting using p62- and L2- (L2-1) specific mAbs. Β-actin was used as loading control. The black asterisk marks an unspecific band. (**b**,**c**) Quantification of Western blot shown in (**b**) was performed for each condition using ImageJ software. The values are given as a mean ± SEM, and the mean for control siRNA-treated cells without bafilomycin A1-addition was set to 100%. Data (*n*= 6–10 out of at least three replicates) were analyzed for significant differences for L2 using non-parametric ANOVA (Kruskal–Wallis test): *p* < 0.0001. Comparison for groups of interest is marked, and *p* values were calculated using unpaired *t* test (*p* <0.0001 for contr.—baf. vs. p62 si—baf.) or Mann–Whitney test (*p* = 0.0002 for contr.—baf. vs. contr. + baf. and p62 si—baf. vs. p62si + baf., and *p* > 0.05 for contr. + baf. vs. p62 si + baf.). Statistical differences in p62 were analyzed using non-parametric ANOVA (Kruskal–Wallis test): *p* = 0.0002. Comparison for groups of interest was calculated using Mann–Whitney test (*p* = 0.0083 for contr. vs. p62 si and *p* = 0.0041 for contr.—baf. vs. contr. + baf.). Mean values ± SEM are displayed below the Western blot duplicates. Note, the protein amounts of p62, the well-known autophagy cargo, increased upon bafilomycin A1 treatment and are used as control for autolysosome inhibition. (**d**,**e**) vDNA accumulates in LC3B-positive structures after p62 knockdown. (**d**) Representative images of vDNA (green) and LC3B (red) co-localization. Control siRNA-transfected cells (contr.) are shown in the upper panel and p62 siRNA-transfected cells (p62 si) in the lower panel. Arrows mark co-localizing vDNA (green) and LC3B (red). Boxes mark the outline of vDNA and PML co-localization, shown in higher magnification. HeLa cells were treated with control or p62-specific siRNA pool for 48 h, incubated with EdU-labeled HPV16 PsV for 8 h and stained with mAb anti-LC3B. During PsVs incubation, cells were treated with bafilomycin A1 (1.25 µM) to prevent autophagic degradation of viral particles. (**e**) Relative co-localization of vDNA and LC3B. vDNA pixels co-localizing with LC3B pixels are given as a mean ± SEM, and the mean for control siRNA-treated cells (contr.) was set to 100%. Data (*n* = 60 out of three replicates) were analyzed for significant differences using Mann–Whitney test; *p* < 0.0001. (**f**,**g**) p62 is required for HPV16 trafficking to PML NBs. (**f**) Representative pictures of vDNA (green) and PML (red) co-localization after p62 knockdown. Cells were treated as in (**d**), incubated with EdU-labeled HPV16 PsVs for 24 h and stained with PML-specific mAb. Control siRNA-transfected cells (contr.) are shown in the upper panel and p62 siRNA-transfected cells (p62 si) are shown in the lower panel. Arrows mark co-localizing vDNA (green) and PML (red). Boxes mark the outline of vDNA and PML co-localization shown in higher magnification. (**g**) Relative co-localization of vDNA and PML. vDNA pixels co-localizing with PML pixels are given as a mean ± SEM, and the mean for control siRNA-treated cells (contr.) was set to 100%. Data (*n* = 57–58 out of three replicates) were analyzed for significant differences using Mann–Whitney test; *p* = 0.0013. (*p* ≤ 0.01 **, *p* ≤ 0.001 ***, not significant n.s.).

**Figure 5 viruses-14-01478-f005:**
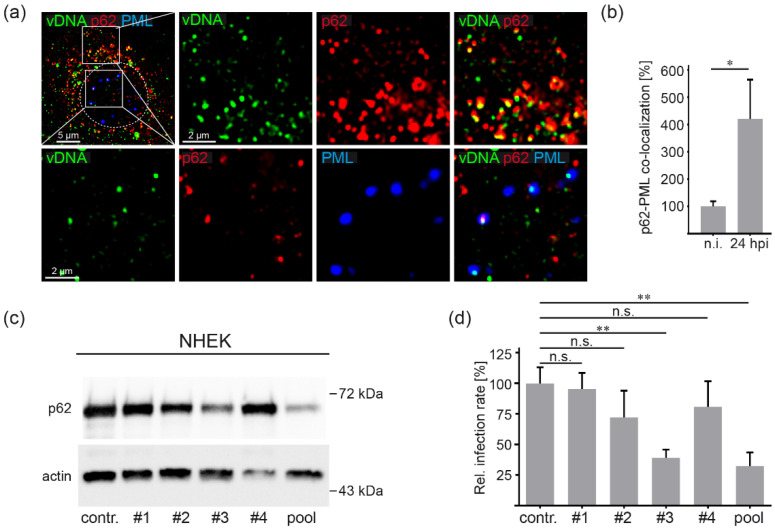
HPV16 PsV infection in primary keratinocytes involves p62. (**a**) Representative pictures of vDNA (green), p62 (red) and PML (blue) co-localization in primary keratinocytes. NHEKs were treated with EdU-labeled HPV16 PsVs for 24 h and stained with p62- and PML-specific mAbs. Pictures show higher magnification of vDNA-p62 co-localization in the cytoplasm (upper panel) or vDNA-p62-PML co-localization in the nucleus (lower panel) as outlined by boxes in the first panel (top left). (**b**) Relative co-localization of p62 and PML in NHEK cells. NHEKs were incubated with EdU-labeled HPV16 PsV for 24 h and stained with p62- and PML-specific mAbs. P62 pixels co-localizing with PML pixels are given as a mean ± SEM, and the mean for non-infected cells (n.i.) was set to 100%. Data (*n* = 57–60 out of three replicates) were analyzed for significance using Mann–Whitney test; *p* = 0.0251. (**c**,**d**) NHEKs were transfected with control (contr.) or p62-specific siRNAs as indicated for 48 h and incubated with HPV16 LCR PsVs for an additional 24 h. (**c**) Knockdown efficiency was analyzed by Western blotting using p62-specific mAb 48 h after siRNA transfection. β-actin was used as loading control. (**d**) Relative infection rates were assessed by luciferase activity and normalized to LDH activity as cell viability control. Infection rate is given as mean ± SEM, and the mean for control siRNA-treated cells was set to 100%. Data (*n* = 7–9 out of three replicates) were analyzed for significance using non-parametric ANOVA (Kruskal–Wallis test): *p* < 0.0036. Comparison for groups of interest is marked, and *p* values were calculated using unpaired *t* test (*p* > 0.05 for contr. vs. siRNA#1, *p* > 0.05 for contr. vs. siRNA#2, *p* > 0.05 for contr. vs. siRNA#4, *p* = 0.0015 for contr. vs. siRNA pool) and Mann–Whitney test (*p* = 0.0055 for contr. vs. siRNA#3). (*p* ≤ 0.05 *, *p* ≤ 0.01 **, not significant n.s.).

**Figure 6 viruses-14-01478-f006:**
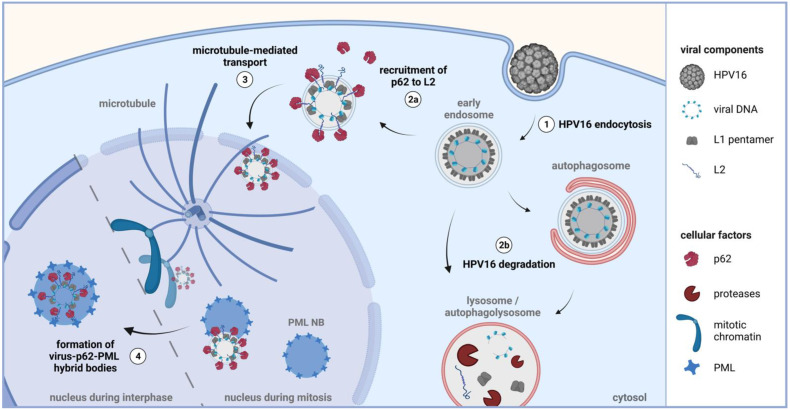
Model of p62 nuclear translocation and formation of virus-p62-PML hybrid bodies during HPV16 entry. (**1**) HPV16 enters its host cell via endocytosis. Maturation of the virus-filled early endosome enables L2 insertion into the endosomal membrane and allows for L2 interaction with cytoplasmic p62 (**2a**). This interaction protects the virus from autophagosomal and/or lysosomal degradation by proteases (**2b**). With the onset of mitosis, associations with microtubules and mitotic chromatin facilitate further transportation of the of virus-filled vesicles into the newly formed nucleus (**3**). Here, HPV16-p62 complexes fuse with re-assembling PML NBs and form virus-associated p62-PML hybrid bodies (**4**), enabling immune evasion and viral gene transcription. Created with BioRender.com (accessed on 31 May 2022).

## Data Availability

Not applicable.

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
