# Peer review of "HPV16 Induces Formation of Virus-p62-PML Hybrid Bodies to Enable Infection"

_viruses, 2022, doi:10.3390/v14071478_

Round 1

Reviewer 1 Report

The manuscript from Linda Schweiger et al., titled “HPV16 induces formation of virus-p62-PML hybrid bodies to enable infection” describes p62 localizes to promyelocytic leukemia nuclear bodies (PML NBs) early upon infection with pseudo-HPV16 viruses. Authors discuss viral particles interact with p62 through L1 and viruses are sequestered into PML NBs and protected them from autophagic degradation. While authors convincingly demonstrate p62 localizes with viral particles into PML NBs following infection, the role of this localization in protecting HPV from autophagic degradation should be more detailed. Based the following considerations, several concerns should be addressed to allow the publication on Viruses journal:

Main points:

1.       In figure 1C authors sustain dots are representative of autophagic vesicles or endosomes. However, to reach this conclusion specific markers should be used. In addition, a proper quantification of p62 and L1 co-localization is necessary.

2.       The conclusion reported in lane 316-317: “These data support the notion of p62 protecting L2 from degradation, most likely via the autophagosomal pathway” is not fully supported by showed results. In figure 4b, authors  how the addition of Bafilomycin in p62 silenced cells increases L2 levels similarly to untreated cells, suggesting p62 is not moduating L2 turnover trough autophagy. In addition, a longer exposition should be added to better appreciate different protein levels between lane 1-2 and 3-4. Moreover, also L1 protein should be checked by WB.

3.       Autophagy is a dynamic process and to better evaluate this cellular mechanism, the analysis should be performed at different time point following infection (7hpi, 24hpi, 48hpi) monitoring its autophagic degradation adding or not Bafilomycin. LC3 WB should be added to monitor the proper autolysosome inhibition; moreover, cellular exposition to Bafilomycin should be reduce from 24hours to few hours (2-4h) to give a picture of autophagy contribution on L2 degradation via autophagy.

4.       In addition, while WB analysis of L2 in figure 4b-c has been performed 24hpi plus Bafilomycin, immunofluorescence in figure 4d-e is 8hpi plus Bafilomycin, and figure 4f-g 24hpi without Bafilomycin. It would be necessary to evaluate similar experimental condition.

5.       Supplementary figure S3 and figure 5d report virus internalization in HeLa cells by immunofluorescence and infection rate on primary keratinocytes following p62 siRNA, respectively. Since effects seem different, it is important to evaluate virus internalization on NHEKs and, viceversa the infection rate on HeLa cells following p62 silencing.

6.       To extensively evaluate the contribution of p62 in reducing virions degradation via autophagy, it would be interesting to evaluate the rate of p62-vDNA-PML normalizing it on cytosolic p62-vDNA co-localization, during infection. This experiment would give an idea about the relevance of the proposed mechanism during infection.

7.       Considering authors conclude L2 is more degraded by autophagy following p62 silencing and, that it is well established p62 is a protein essential in autophagy activity; it is important to explore or, at least discuss, other p62-like protein able to mediate virions degradation by autophagy.

Minor points:

1.       In figure 1a, the y-axis is refered to peptides or ptoein aboundances? Specify in the graph.

2.       Lane 282: “(Figure 3a). and were used in the HPV16”, remove the full stop.

3.       Lane 300: Add the full stop following “[42]”.

Author Response

Point by point response to Reviewer 1:

Main points:

  1. In figure 1C authors sustain dots are representative of autophagic vesicles or endosomes. However, to reach this conclusion specific markers should be used. In addition, a proper quantification of p62 and L1 co-localization is necessary.

First, we agree with the reviewer and removed the statement “dots are representative of autophagic vesicles or endosomes” at this point of the manuscript as figure 1 focusses on endosomes.

Second, we now show the complete list with quantification of endosomal proteome (new suppl. figure - Figure S1). Here, endosomal markers are found (e.g., Rab5a, also shown in Fig. 1b) while no LC3 was detected in endosomal fractions.

Third, we over-expressed LC3B-GFP, as specific marker for autophagosomes and detected slight overlap of capsid protein L1 staining and LC3B-GFP signal. When the co-localization of LC3B-GFP and L1 was analyzed at different time points post infection, no changes in co-localization were detectable (new suppl. figure - Figure S5). These data suggest a continuous flux of incoming PsVs via the autophagy pathway. Therefore, we investigated specific time points under steady state conditions or blocked autophagy in this study. E.g., quantification of a virus component (vDNA) and endogenous LC3B was detectable in the presence of bafilomycin A1 and is shown in Figure 4e. Here, we have incubated the cells for 8 hours with PsVs to measure the maximum amount of cytoplasmic vDNA.

  1. The conclusion reported in lane 316-317: “These data support the notion of p62 protecting L2 from degradation, most likely via the autophagosomal pathway” is not fully supported by showed results. In figure 4b, authors show the addition of Bafilomycin in p62 silenced cells increases L2 levels similarly to untreated cells, suggesting p62 is not modulating L2 turnover trough autophagy. In addition, a longer exposition should be added to better appreciate different protein levels between lane 1-2 and 3-4. Moreover, also L1 protein should be checked by WB. 

Under steady state conditions (in the absence of Bafilomycin A1) the L2 levels are reduced in the absence of p62, while under Bafilomycin A1 treatment the L2 levels were comparable. This suggests that p62 modulates L2 turnover through autophagy as the autophagic flux is blocked independent of the p62 presence or absence. As Bafilomycin A1 is also able to block lysosomal degradation, we toned down the statements made in the text. Most of L1 is lost during viral entry and degraded via lysosomal degradation, so we focused these analyzes on L2, which accompanies viral DNA into the nucleus. We have now added a sentence that explains our focus.

In addition, the shorter exposure is now replaced by a longer exposure (4b) and relative band intensity of the L2 bands are added below the bands in addition to the summary shown in Fig 4C.

  1. Autophagy is a dynamic process and to better evaluate this cellular mechanism, the analysis should be performed at different time point following infection (7hpi, 24hpi, 48hpi) monitoring its autophagic degradation adding or not Bafilomycin. LC3 WB should be added to monitor the proper autolysosome inhibition; moreover, cellular exposition to Bafilomycin should be reduce from 24hours to few hours (2-4h) to give a picture of autophagy contribution on L2 degradation via autophagy. 

We have now added a new suppl. Fig. S5 showing analysis of LC3B-GFP-L1 co-localization during the time course of infection at 5, 7, 9 24 hours post infection (see also point 1). In addition, we quantified p62 WB to monitor the proper autolysosome inhibition (Fig. 4b and quantification) and added the sentence „Note, the protein amounts of p62, the well-known autophagy cargo, increased upon bafilomycin A1 treatment and is used as control for autolysosome inhibition. “

  1. In addition, while WB analysis of L2 in figure 4b-c has been performed 24hpi plus Bafilomycin, immunofluorescence in figure 4d-e is 8hpi plus Bafilomycin, and figure 4f-g 24hpi without Bafilomycin. It would be necessary to evaluate similar experimental condition. 

We apologize for the confusion and for not explaining this point better. Through our many years of experience in the field of HPV cell entry, we know that a maximum of viral particles can be found in endosomes at 8 hpi (e.g., Popa et al., 2015; https://doi.org/10.1371/journal.ppat.1004699). These particles undergo capsid disassembly and degradation (Spoden et al., 2008; Gräßel et al., 2016; Cerqueira et al., 2015, DOI:https://doi.org/10.1128/JVI.00234-15) and in the cell nucleus co-localizing with PML at 24 hours post-infection (e.g., Bund et al., 2014 Guion et al., 2019, https://doi.org/10.1371/journal.ppat.1007590). We have therefore chosen these durations of PsV incubation for the different analyses. Corresponding explanations and references are now included into the manuscript.

  1. Supplementary figure S3 and figure 5d report virus internalization in HeLa cells by immunofluorescence and infection rate on primary keratinocytes following p62 siRNA, respectively. Since effects seem different, it is important to evaluate virus internalization on NHEKs and, vice versa the infection rate on HeLa cells following p62 silencing. 

      HPV16 PsV infection in both HeLa and NHEK cells was already shown in Figures 3b and 5d, respectively, of the original version of the manuscript. Figure S3 is just one of the multiple controls that were additionally performed for HeLa showing that PsV endocytosis is not affected by p62 KD. In NHEKs, we reproduced key findings and found comparable results to HeLa. A minor difference is the lower siRNA or knockdown efficiency in NHEK as primary keratinocytes are less transfectable than HeLas. In the new Figure 5d, we have replaced infection analysis using only the pool of p62 siRNA by analyzing the effect of all 4 siRNAs and the pool thereof. In addition, we now used PsVs that contain a reporter gene plasmid with the marker gene luciferase under control of the HPV16 long control region containing the HPV16 viral early promoter. This approach combines the HPV16 target cells (primary NHEK) with an HPV16 PsVs system that mimics the natural infection. Our findings demonstrate that PsV infection rates correlated with reduction of p62 protein levels caused by treatment with the different siRNAs and indicates that p62 plays a pro-viral role in natural infections. Further, data support the relevance of the conclusions drawn from the findings obtained in HeLa experiments. The functional importance of p62 for HPV16 LCR PsV infection in primary keratinocytes is now shown in Figure 5d along with the p62 WB for all siRNAs on Figure 5c.

  1. To extensively evaluate the contribution of p62 in reducing virions degradation via autophagy, it would be interesting to evaluate the rate of p62-vDNA-PML normalizing it on cytosolic p62-vDNA co-localization, during infection. This experiment would give an idea about the relevance of the proposed mechanism during infection. 

The way we understand the reviewer’s question, it relates to two different mechanisms proposed in the manuscript: 1) p62 facilitates virus trafficking into PML bodies and 2) p62 protects L2 from degradation. We think that these two mechanisms depend on each other. Blockage of trafficking into the nucleus by p62 depletion might cause increased degradation. Whether p62 protects HPV16 from degradation by facilitating translocation into the nucleus is therefore a very interesting question but for now remains to be determined in a follow up study.

  1. Considering authors conclude L2 is more degraded by autophagy following p62 silencing and, that it is well established p62 is a protein essential in autophagy activity; it is important to explore or, at least discuss, other p62-like protein able to mediate virions degradation by autophagy.

We agree with the reviewer and discussed other p62-like proteins which are able to mediate virion degradation by autophagy – especially in the absence of p62. On the other hand, p62 binds to ubiquitinated proteins and has multiple reported functions as summarized in Yoshinori Katsuragi et al., (FEBS J 2015; DOI: 10.1111/febs.13540).

Minor points:

  1. In figure 1a, the y-axis is referred to peptides or protein abundances? Specify in the graph.

Parts per million (ppm) values are calculated at the protein level. The amount of each protein is determined using the 3 "best ionizing" peptides, which are then compared to the total protein in the sample. Explanations are now included into the figure legends (Fig. 1a and S1) and is specified in the graph.

  1. Lane 282: “(Figure 3a). and were used in the HPV16”, remove the full stop.

Has been corrected.

  1. Lane 300: Add the full stop following “[42]”.

Has been corrected.

Reviewer 2 Report

Comments:

- Line 31: Provial should proviral.

-Lines 94 to 98: HPV16 pseudoviruses constructs and prodcution should be presented in more details. For such a critical element, the description should be clear and detailed rather than directing the readers to an array of publication.

- Material and methods should include clear heading to improve reader experience.

- Line 184: How is the reanalysis of the mass spec data different from the analysis performed in reference 35? Has the analysis fundamentally changed or is p62 just another hit gene? Is the complete list available for readers in the previous publication? These should be clearly stated/explained.

-Figure 1B: LC3I and LC3II should also be probed to better define the state of autophagosome.

-Figure 1C: Could the authors show staining with LC3II as well?

- Figure 1D/E: How do the authors explain that L2 and p62 interact in the absence of the viral capsid? To be able to say that p62 interact with the cytoplasmic part of L2 (lines 203 - 204) authors need to perform the co-IP experiment with a L2 truncation mutant lacking the site and show that interaction is abrogated.

-Figure 3A: Does the pool of siRNA include si#4?

-Figure 3B: Luciferase data is not very convincing. Can the authors show decrease viral replication through qPCR of  viral gene mRNA expression or viral copy number?

-Figure 4A and C: Statistical test should be one-way ANOVA, not t-test...

-Figure 4B: It is not clear why there are 2 samples for each categories. Please explain. In addition, controls are missing in this experiment: Please perform immunoblotting of LC3I/II to confirm impact on autophagy and authors need to add non-infected cells controls for both mock and BafA1 conditions. All together that panel should probably be redone. Finally please add the quantification relative intensity of the L2 bands below the bands in adition to the summary (Fig 4C). Changes are very minimal and all of this would help be more convincing.

-Figure 5C is not acceptable. Why is p62 siRNA and the control siRNA from different panels! They should be run side by side! How many times has this data been repeated?

Author Response

Point by point response to reviewer 2:

We would like to thank reviewer 2 for his/her comments that helped us to improve the manuscript.

- Line 31: Provial should proviral.

„proviral“ has been corrected in the revised manuscript.

-Lines 94 to 98: HPV16 pseudoviruses constructs and production should be presented in more details. For such a critical element, the description should be clear and detailed rather than directing the readers to an array of publication.

We agree with the reviewer and included HPV16 pseudoviruses constructs and production.

- Material and methods should include clear heading to improve reader experience.

As requested, Material and methods are now structured by clear heading.

- Line 184: How is the reanalysis of the mass spec data different from the analysis performed in reference 35? Has the analysis fundamentally changed or is p62 just another hit gene? Is the complete list available for readers in the previous publication? These should be clearly stated/explained.

p62 is indeed another hit of the described mass spec screen published in 35. As the complete list was not available for the readers, we now added the complete list as new suppl. fig S1. In addition, we added the extended experimental procedure to the supplementary methods and a paragraph to the main manuscript.

-Figure 1B: LC3I and LC3II should also be probed to better define the state of autophagosome.

In experiments shown in Fig. 1B, we have isolated endosomes and were unable to detect LC3 in any of the mass spec analyses. For clarification, we have now included the sentence “The autophagy adaptor p62/sequestosome-1 was identified as a strongly enriched protein at four and seven hours post PsVs infection (hpi) (Figure 1a, Figure S1), while no autophagosome marker (e.g. LC3) was detected (Figure S1)“.-Figure 1C: Could the authors show staining with LC3II as well?

We have now analyzed the co-localization between incoming PsVs (L1) and LC3B-GFP. As we have found very few events of co-localization with endogenous LC3B, we overexpressed LC3B-GFP and detected slight overlap of capsid protein L1 staining and LC3B-GFP signal. When the co-localization of LC3-GFP and L1 was analyzed at different times of infection, no changes in LC3-GFP-L1 co-localization were detectable (new supplementary figure - Figure S5). These data suggest a continuous flux of incoming PsVs via the autophagy pathway. In order to keep the text flowing, we connected this experiment to Figure 4 in which data for autophagy are shown.

- Figure 1D/E: How do the authors explain that L2 and p62 interact in the absence of the viral capsid? To be able to say that p62 interact with the cytoplasmic part of L2 (lines 203 - 204) authors need to perform the co-IP experiment with a L2 truncation mutant lacking the site and show that interaction is abrogated.

We apologize for the confusion, interaction of p62 with L2 in the absence of the capsid is impossible in infection assays as there is no infection without L1. Therefore, we transfected the cells with an L2-expression plasmid. We additionally apologize for not describing the plasmid in the original version. It is now described in Materials and Methods and over-expression is mentioned in the text.

 In co-IP experiments of these cells, we find strong co-precipitation of endogenous p62 when L2 is precipitated indicating complex formation of these two proteins and suggesting that interaction of incoming PsV is mediated via p62-L2 interaction. This is supported by multiple earlier studies showing that L1 is not accessible for cytoplasmic proteins as it remains within a transport vesicle during virus entry. During this process, only L2 becomes trans-membranous and interacts with proteins in the cytosol.

We agree with the reviewer and removed the statement that interaction is mediated via the C-terminus throughout the manuscript.

-Figure 3A: Does the pool of siRNA include si#4?

Indeed, the pool includes siRNA #4 as better described now in Materials and Methods “… or a pool of all four single siRNAs using RNAiMAX”.

-Figure 3B: Luciferase data is not very convincing. Can the authors show decrease viral replication through qPCR of viral gene mRNA expression or viral copy number?

-Figure 4A and C: Statistical test should be one-way ANOVA, not t-test.

To our knowledge, "ordinary one-way ANOVA" tests for significant differences among the means of more than 2 groups of samples with the assumption of normally distributed data. However, this test alone does not test for significant changes between means of each of the groups. Therefore, we now included calculated p-values from parametric or non-parametric ANOVA tests for normally and non-normally distributed data, respectively, in addition to statistical tests among different groups.

-Figure 4B: It is not clear why there are 2 samples for each category. Please explain. In addition, controls are missing in this experiment: Please perform immunoblotting of LC3I/II to confirm impact on autophagy and authors need to add non-infected cells controls for both mock and BafA1 conditions. All together that panel should probably be redone. Finally, please add the quantification relative intensity of the L2 bands below the bands in addition to the summary (Fig 4C). Changes are very minimal and all of this would help be more convincing.

Two replicates were loaded to demonstrate reproducibility and to show the uncropped image. Relative intensities of the L2 bands below the WB bands are now shown in addition to the summary. Moreover, we quantified p62 bands as a well-known autophagy cargo. Comparison for contr. - baf. vs. contr. + baf. p = 0.0041) serves as control for blocked autophagy. We included the following sentence to the legend „Note, the protein amounts of p62, the well-known autophagy cargo, increased upon bafilomycin A1 treatment and is used as control for autolysosome inhibition. “

-Figure 5C is not acceptable. Why is p62 siRNA and the control siRNA from different panels! They should be run side by side! How many times has this data been repeated?

The whole WB was added as supplementary file to the journal and is now included into the main manuscript. Repetitions are indicated in the figure legend (n=7-9 out of three replicates).

Round 2

Reviewer 2 Report

Authors made efforts to address most of reviewers comments.